# SOS-Inducing Drugs Trigger Nucleic Acid Release and Biofilm Formation in Gram-Negative Bacteria

**DOI:** 10.3390/biom14030321

**Published:** 2024-03-08

**Authors:** Peter Demjanenko, Sam Zheng, John K. Crane

**Affiliations:** 1Jacobs School of Medicine and Biomedical Sciences, University at Buffalo, Room 317 Biomedical Research Bldg., 3435 Main St., Buffalo, NY 14214, USA; pdemjane@buffalo.edu (P.D.); puzheng@buffalo.edu (S.Z.); 2Division of Infectious Diseases, Department of Medicine, Jacobs School of Medicine and Biomedical Sciences, University at Buffalo, Room 317 Biomedical Research Bldg., 3435 Main St., Buffalo, NY 14214, USA

**Keywords:** SOS response, single-stranded DNA, RNA, nitric oxide, zinc, dequalinium, *Escherichia coli*, *Klebsiella pneumoniae*, *Salmonella enterica*, *Shigella sonnei*, *Pseudomonas aeruginosa*

## Abstract

Our laboratory recently reported that induction of the SOS response, triggered by SOS-inducing drugs, was accompanied by a large release of DNA from enteric bacteria. The SOS response release had not previously been reported to include release of extracellular DNA from bacterial cells. We followed up on those observations in this current study and found that not just double-stranded DNA was being released, but also single-stranded DNA, RNA, and protein. SOS-inducing drugs also triggered formation of biofilm at the air–fluid interface on glass, and the biofilms contained DNA. We extended our study to test whether inhibitors of the SOS response would block DNA release and found that SOS inhibitors, including zinc salts, nitric oxide donors, and dequalinium, inhibited SOS-induced DNA release. The understanding that SOS-induced DNA release is associated with formation of biofilms increases our appreciation of the role of the SOS response in pathogenesis, as well as in emergence of new antibiotic resistance. Our findings with SOS inhibitors also suggest that regimens might be devised that could block the deleterious effects of the SOS response, at least temporarily, when this is desired.

## 1. Introduction

The SOS response is a well-studied stress response pathway in Gram-negative and Gram-positive bacteria, which has been especially well-studied in *Escherichia coli*. The term “SOS Repair” was coined by Miroslav Radman in 1975 as an example of a DNA repair pathway that also introduces new mutations into the genome. The idea that DNA could be repaired while also introducing new mutations was a foreign concept for many because it seemed to be a contradiction or an oxymoron. But this idea has indeed been borne out, and studies in recent years have highlighted the role of the SOS response in the generation of new antibiotic resistance [1]. Despite the long pedigree of research on the SOS response, this pathway continues to produce new and unexpected findings. One recurring finding is that the SOS response is triggered by drugs other than antibiotics, including anti-cancer drugs, antivirals, and antidepressants. Another theme that has emerged over the last five years is that other bacterial stress response pathways, including the oxidative stress response, the generalized stress response, and the stringent response, can all prolong and intensify events in the SOS pathway, including hypermutation. These pathways interact and interconnect with one another in interesting and important ways, and they can be blocked by different inhibitors, suggesting the future possibility of combination therapy to prevent hypermutation and antibiotic resistance.

In our previous study, we discovered that drug-induced SOS response triggered a large release of DNA into the extracellular medium in *Enterobacter cloacae* and *E. coli* [2]. DNA release had not previously been reported as a part of the SOS response. The extracellular DNA was entangled amongst large clumps of bacteria, which were also elongated as a result of the SOS response, a process also called filamentation. In this current study we show that single-stranded DNA (ssDNA) and RNA are also released into the extracellular medium in response to the SOS response. This is intriguing because ssDNA is the form of DNA recognized by the RecA protein during the formation of the RecA nucleofilament, and RecA protein is also released into the supernatant during the SOS response [3].

Recent reports in the literature indicated that DNA damage responses in bacteria could also include SOS-independent pathways [4]. We realized that our previous report [2] had not definitively determined whether the DNA release we observed was strictly SOS-dependent. We addressed this question using a ΔrecA mutant and found that the DNA release was indeed dependent on RecA, a key early signal transducing molecule in the SOS response. We next examined if treatment with SOS-inducing drugs triggered DNA release in bacteria other than *E. coli* and Enterobacter, and found that DNA release was observed in most, but not all, strains tested. Bacterial capsule appears to be one factor that impedes release of DNA into the supernatant medium), but there may be others as well.

In this study, we also extended our observations about bacterial clumping and showed that SOS inducers triggered formation of robust biofilms on inanimate surfaces, such as glass. The SOS response had previously been implicated as a trigger of biofilm formation, but the role of DNA in the biofilms was not appreciated [5]. Bacterial biofilms have been recognized as sanctuary sites in which emergence of antimicrobial resistance occurs [6].

We previously showed that the SOS pathway is blocked by zinc salts and by nitric oxide donors, and both of these inhibitors acted by preventing the RecA protein from binding to ssDNA, a key early event in the SOS response [7,8]. In this study, we tested these inhibitors for their ability to block release of nucleic acid from bacterial cells. Dequalinium, a quaternary ammonium compound, has recently been reported to inhibit the generalized stress response, mediated by transcription factor σ^s^ [9]. We tested dequalinium for its ability to block DNA release from bacteria and found it to be promising.

In summary, our previous report on SOS-induced dsDNA release was incomplete or misleading because we failed to recognize that ssDNA and RNA are also released in large amounts, as well as protein. Extracellular DNA is incorporated into biofilms, where it could play a role in pathogenesis as well as in emergence of antibiotic resistance.

## 2. Materials and Methods

### 2.1. Bacterial Strains Used, and Microbial Growth Conditions

Bacterial strains used are shown in Figure 1. Bacteria were grown exactly as described by Crane and Catanzaro [2]. SOS inducers were added at the 1.5 h time point of subculture, and nucleic acids and biofilms were measured at 4 or 4.5 h.

### 2.2. Reagents Used

Most of the reagents used were the same as reported by Crane and Catanzaro. Dequalinium HCl, mitomycin C, azithromycin, and S-nitroso-acetyl-penicillamine (SNAP) were from Cayman Chem, Ann Arbor, MI. Polymyxin B, bovine serum albumin, and Gel Green were from Millipore-Sigma (St. Louis, MO, USA). RecA protein was from New England Biolabs, and anti-dsDNA antibody was from Abbomax, San Jose, CA, USA. Dequalinium is quite insoluble in water. Therefore, a 100× stock of dequalinium in an Eppendorf tube was warmed before use by soaking in a small beaker of water at 95 °C for 5 min; after this, the dequalinium was resuspended by vigorously pipetting up and down, then quickly added to the recipient bacterial suspension.

### 2.3. Measurement of Nucleic Acids by Fluorometry

Double-stranded DNA, single-stranded DNA, and RNA were measured using the Qubit Flex fluorometer from Thermo-Fisher, (Grand Island, NY, USA) using the reagents supplied with the appropriate assay kits. In later experiments, we used the assay kits from Biotium (Fremont, CA, USA) for fluorometry instead, and these also performed well.

### 2.4. Protein Assays

Protein was measured by the method of Bradford, using the protein assay kit from Bio-Rad (Carlsbad, CA, USA) [10]. The color generated from Coomassie Blue binding was measured at 595 nm using a 96-well plate reader.

### 2.5. Detection of Biofilms

Biofilms could be seen with the naked eye on glass test tubes, where they formed at the air–fluid interface. For quantitation of DNA in biofilms, we grew the strains in 16 × 100 mm borosilicate glass test tubes containing glass microscope slides. We cut the microscope slides vertically using a glass cutter, and then sterilized the cut slides by autoclaving. The biofilms were allowed to air dry, rinsed twice with phosphate-buffered saline, then stained with SybrSafe DNA stain (Thermo-Fisher/Invitrogen, Grand Island, NY, USA) or Gel Green at a dilution of 1:5000, twice the concentration usually used to visualize DNA in agarose gels. The biofilms were then imaged using a Bio-Rad Gel Doc EZ fluorescence imager, using the settings for SybrSafe dye. In order to visualize the DNA fluorescence in the biofilms, the manual settings of the Gel Doc EZ instrument were used, with an exposure time of 12 s. Images of the biofilms were quantitated using the Un-Scan-It Gel program (Silk Scientific, Orem, UT, USA).

### 2.6. DNA Analysis by Agarose Gels

DNA gels were run as described by Crane and Catanzaro [2].

### 2.7. Data Analysis

Data shown are in the form of means ± standard deviations. Graphs and statistical testing were performed using GraphPad Prism for MacIntosh (San Diego, CA, USA). A *p* value of ≤0.05 was chosen for significance, as is standard in biomedical research.

## 3. Results

In our previous study [2], we relied on spectrophotometric measurement of DNA released from bacteria treated with SOS-inducing drugs. In the current study, we converted to fluorometric measurements of nucleic acids so that we could accurately distinguish between dsDNA, ssDNA, and RNA. In light of recent reports on bacterial DNA damage pathways that are SOS-independent [4], we realized that our previous findings, while suggestive, did not prove that the SOS response was responsible for the DNA release. Therefore, we compared basal and drug-induced DNA release in *E. coli* strain EDL933, an STEC strain, with its ΔrecA mutant, EDL933R. Figure 1A shows that the SOS-inducing drugs bleomycin and mitomycin C triggered release of dsDNA in the wild-type EDL933 but not the EDL933R mutant. For thoroughness, we also measured ssDNA release. To our surprise, the amount of ssDNA released by bleomycin and mitomycin C greatly exceeded dsDNA by a factor of about five (Figure 1B). However, release of ssDNA was still completely abolished in the EDL933R mutant. Since RecA is the key molecule involved in recognition of DNA damage and for initiating the SOS response, we believe that the findings in Figure 1 indicate that nucleic acid release is mediated via the SOS response. We searched the literature and failed to find definite reports that the SOS response triggered release of DNA or any other nucleic acid. One faint hint in the literature was that DNA in extracellular biofilms resembled the cruciform structures known as Holliday junctions [11], which can form after DNA damage, but this report did not specifically mention the SOS response.

Our previous study focused on DNA release from *E. coli* and *Enterobacter cloacae*. For the sake of the generalizability of our findings, we wondered if SOS-inducing drugs triggered DNA release from other Gram-negative bacteria. Figure 2A shows that drugs such as ciprofloxacin, zidovudine, bleomycin, and mitomycin C caused a strong release of DNA from *Shigella sonnei*, but almost none from *Salmonella enterica. S. enterica* has a well-studied SOS pathway, so the findings in Figure 2A were unexpected. A recent report by Merida-Floriano et al. suggests that either a more rapid cell death in Salmonella or differences in SOS events downstream of LexA cleavage might explain the differences compared to *E. coli* [12].

**Figure 2 biomolecules-14-00321-f002:**
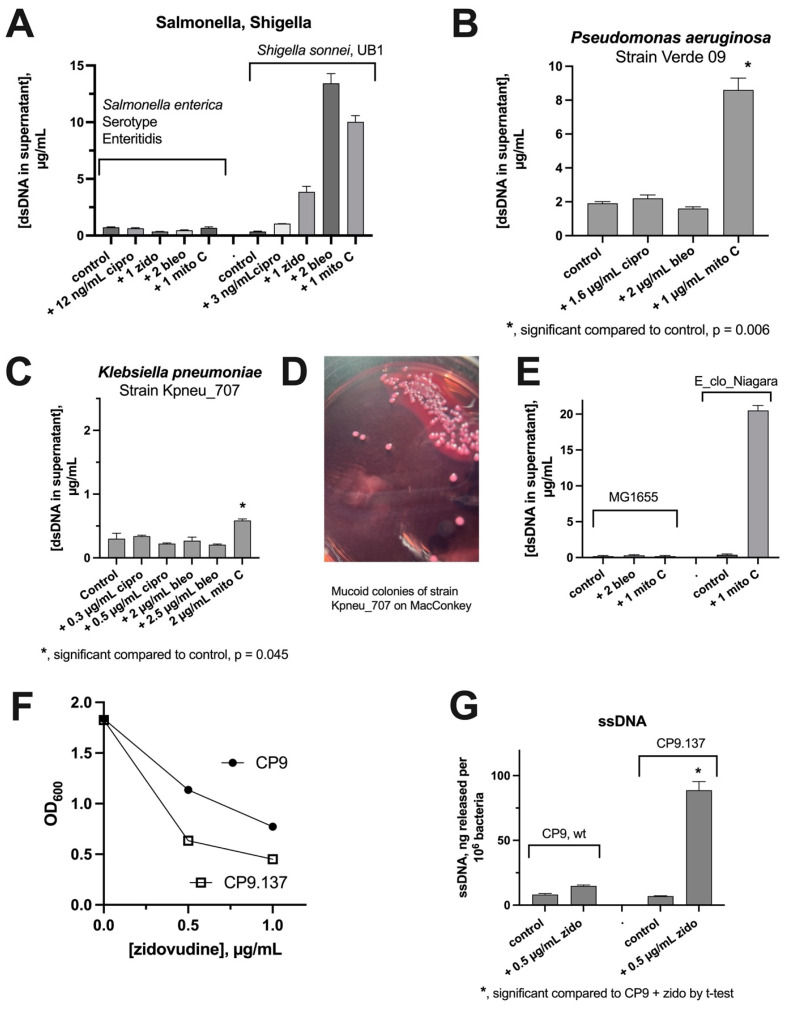
Comparison of various bacterial strains for SOS-induced DNA release. Various strains of Gram-negative bacteria, as described in Table 1, were tested and compared with regard to dsDNA release after treatment with SOS-inducing drugs. Panels (**A**–**C**,**E**) show release of dsDNA. Panel (**D**): encapsulated strain of Klebsiella; Panel (**F**), CP9.137 is hypersusceptible to zidovudine. Panel (**G**): release of ssDNA by CP9 and CP9.137, normalized to viable bacterial counts. Cipro, ciprofloxacin; zido, zidovudine.

**Table 1 biomolecules-14-00321-t001:** Bacterial strains used.

Bacterial Strain	Pathotype/Serotype	Comments	Reference(s)
*E. coli* Strains			
EDL933	Shiga-toxigenic *E. coli* (STEC), wt	Stx1+, Stx2+	[13,14]
EDL933R	ΔrecA mutant of EDL933		[15]
CP9	Wild-type ExPEc		[16]
CP9.137	Deficient in K54 capsule	Isogenic derivative of CP9	[17]
E2348/69	Enteropathogenic *E. coli*	Classic human EPEC, + bundle-forming pilus	[18,19]
Iceberg	ExPEc, clinical isolate	Bloodstream isolate, co-infection with COVID-19 Omicron	
MG1655	K-12 Laboratory strain		[20]
Non-*E. coli* Strains			
*Enterobacter cloacae* Strain E_clo_Niagara	Clinical isolate, bacteremia, ESBL	Donor strain for horizontal transfer of CTX-M plasmid	[2,21]
*Klebsiella pneumoniae*, K_pneu_707	Clinical isolate, bacteremic pneumonia	Capsule+	[7]
*Klebsiella aerogenes,* previously *Enterobacter aerogenes*	Clinical isolate	Central line-associated blood stream infection (CLABSI)	
*Salmonella enterica*	Serotype Enteritidis, Clinical isolate	Fecal specimen, gastroenteritis	
*Shigella sonnei*, strain UB1	Clinical isolate	Dysentery, in traveler from Cuba	
*Pseudomonas aeruginosa*, strain Verde 09	Clinical isolate	Respiratory isolate, green pigment	

Figure 2B tested if DNA release could be detected from *Pseudomonas aeruginosa*. Mitomycin C, but not the other inducers, triggered dsDNA release from this strain. Figure 2C shows an experiment testing DNA release from *Klebsiella pneumoniae*. Mitomycin C produced a release of DNA that reached statistical significance, but the amount of DNA was so low that this would seem to be of dubious clinical or biological significance. The Klebsiella strain we used was a mucoid strain (Figure 2D). We wondered if capsule inhibited SOS-induced DNA release. Figure 2E shows the lack of DNA release from *E. coli* MG1655, which expresses K-12 capsule, increasing our suspicion that capsule might interfere with DNA release. To test this more rigorously, we compared SOS-induced DNA release from a wild-type ExPEc strain, CP9, with its isogenic mutant, CP9.137, lacking K54 capsule [17]. CP9 and CP9.137 were much less responsive to mitomycin C than other strains, but released DNA in response to zidovudine. CP9.137 was hypersusceptible to zidovudine (Figure 2F). When DNA release was normalized to viable counts of bacteria, the capsule-deficient mutant showed zidovudine-induced DNA release that was six times higher than the wild-type strain (Figure 2G). Figure 2C–G indicates that bacterial capsule restricts the release of DNA into the supernatant medium. The polysaccharide capsule of *Streptococcus pneumoniae* restricted the release of extracellular DNA in that microbe as well [22,23].

A common refrain from skeptics of the SOS response [24] is that all antibiotics trigger a common pathway of oxidative stress in bacteria and, by extrapolation, that all drugs with antibacterial effects should trigger DNA release. But other authors have called this conclusion into question [25], and others have pointed out that certain antibiotics lack the ability to induce the SOS response and can even inhibit it [26]. We tested if non-SOS-inducing antibiotics triggered DNA release in *E. cloacae*, which has a robust DNA release response, as reported in [2]. Figure 3A shows that chloramphenicol induced only a small dsDNA release, less than 2 µg/mL DNA, and the DNA release decreased further at concentrations of chloramphenicol greater than the MIC. Figure 3A, Right *Y*-axis, shows that chloramphenicol inhibited bacterial growth by 86% at these concentrations, as measured by culture turbidity. Figure 3B shows that azithromycin also failed to cause dsDNA release, while the positive control drug, mitomycin C, did so strongly. Figure 3C shows that polymyxin B, considered a lytic antibiotic due to its ability to bind to lipopolysaccharide (LPS), triggered a small release of dsDNA that was statistically significant, but small compared to bona fide SOS-inducing drugs. Figure 3D shows that aztreonam, a beta-lactam antibiotic that fails to induce the SOS response [2], also failed to trigger DNA release. An additional point to emphasize is that SOS-inducing drugs trigger the maximum release of DNA at concentrations below the MIC, i.e., at sublethal or sub-inhibitory concentrations, while the non-SOS-inducing drugs shown in Figure 3 failed to cause substantial DNA release at any concentrations tested, even those several times higher than the MIC.

Several of the drugs studied in Figure 3A–D, and the non-SOS-inducing drugs studied by Berger [26], were bacteriostatic. We therefore tested bacteriocidal drugs to see if they would trigger DNA release. Zinc pyrithione (ZPT) is a potent bacteriocidal drug that fails to trigger SOS activation [21]. ZPT inhibited bacterial growth (Figure 3E) but failed to induce any DNA release (Figure 3F). Formaldehyde also strongly inhibited growth of Enterobacter, but failed to induce DNA release (Figure 3G,H). Figure 3 shows that drugs that are not SOS inducers, whether bacteriostatic or bacteriocidal, generally did not trigger DNA release. The exception was polymyxin B, which releases intracellular contents via its ability to permeabilize bacterial cells (Figure 3C).

In the course of performing the experiments shown in Figure 1, Figure 2 and Figure 3, we noted that treatment of bacteria with SOS-inducing drugs often induced the formation of biofilm on glass test tubes, at the air–fluid interface of tubes shaken at 300 rpm (Figure 4A). However, these biofilms were not seen when the bacterial strains were grown in polypropylene test tubes. In order to quantitate the biofilms, we tested if biofilms would form on glass microscope slides and found that they would, at least with some strains, such as E_clo_Niagara. We rinsed the slides, allowed them to air dry, and then stained the glass slides with SybrSafe DNA stain. We could then quantitate the amount of dsDNA in the biofilms using a fluorescence gel scanner. The fluorescence of the SybrSafe stain is shown in Figure 4B. Since the bacteria were grown with shaking, the air–fluid interface and the biofilm are in the V-shape of the vortex created around the slide. Figure 4C shows that mitomycin C, the drug that triggered the greatest release of dsDNA in this strain, also increased the amount of DNA in the biofilm.

Single-stranded DNA was released from bacteria by SOS inducers (Figure 1B), and ssDNA is bound by RecA protein. Indeed, release of ssDNA exceeded that of dsDNA by a wide margin (Table 2). Therefore, we next tested if addition of exogenous RecA protein had any effect on biofilm formation in E_clo_Niagara. Figure 4D shows the fluorescence signal from Gel Green, a dye that stains dsDNA, in response to increasing concentrations of RecA protein. RecA protein markedly stimulated the amount of dsDNA in the biofilms at the air–fluid interface (Figure 4D,E). The stimulatory effects of RecA were not observed with two other proteins, bovine serum albumin (BSA) and anti-dsDNA antibody. The biofilms also contained protein, however, as the biofilms also stained with protein stains such as Congo Red (Figure 4F) and with Crystal Violet. Kaushik et al. recently reviewed the role of the SOS response in biofilm formation [27], but did not identify extracellular DNA as an actual component of the biofilm itself. The findings of Figure 4 show that the effects of extracellular DNA surpass promotion of bacterial aggregation, as we previously noted [2], but also extend to formation of biofilms on abiotic surfaces.

### 3.1. Inhibitory Effects of Zinc on Nucleic Acid Release

Several authors and laboratories have noted the potential of SOS inhibitors to be used to prevent emergence of antibiotic resistance or even reverse pre-existing drug resistance [9,28,29,30,31]. We tested if any of the inhibitors of the SOS response that we had identified would also block release of extracellular DNA and nucleic acids. Figure 5 shows the effects of two divalent transition metals, zinc and copper, on SOS-induced DNA release. Figure 5A,B shows that zinc acetate inhibited release of dsDNA and ssDNA from E_clo_Niagara, at the same concentrations previously observed to inhibit RecA expression, hypermutation, and horizontal gene transfer [7,21]. In the interest of thoroughness, we also tested if RNA might be released by SOS-inducing drugs and found that it was (Figure 5C). Drug-induced RNA release was also inhibited by zinc acetate. Figure 5D shows the inhibitory effect of copper sulfate on release of dsDNA. Copper’s effects reach a plateau, however, beyond which higher concentrations of CuSO_4_ fail to cause greater inhibition (compare Figure 5A,D). Nickel chloride failed to inhibit DNA release in this assay.

Figure 5E shows the effects of zinc on the size and patterns of dsDNA released from E_clo_Niagara. The dsDNA release by mitomycin C is seen in Lanes 5 and 6, with the characteristic “W-shaped” artifact seen with mitomycin as the inducer. Zinc appeared to fully block the lower molecular weight DNA bands (400 to 700 bp) in Lanes 7 and 8, while the larger DNA bands were only partially inhibited by zinc. These findings show that zinc affects DNA release qualitatively as well as quantitatively in enteric bacteria.

### 3.2. Inhibitory Effects of Nitric Oxide on Nucleic Acid Release

Nitric oxide donors also inhibit the SOS response, and do so by blocking the ability of the RecA protein to bind to ssDNA [8]. Dequalinium hydrochloride is not an SOS inhibitor per se, but, by inhibiting the generalized stress response in bacteria, it can diminish the effects of SOS inducers [9]. Our experiments testing these inhibitors on E_clo_Niagara are shown in Figure 6. As shown in Figure 6A–C, the nitric oxide donor S-nitroso-acetyl-penicillamine (SNAP) inhibited mitomycin-induced release of dsDNA, ssDNA, and RNA, respectively. Of these three nucleic acids, SNAP appeared to be most potent and complete in its ability to block release of dsDNA and ssDNA, with 8 mM SNAP achieving >70% inhibition of DNA release. Inhibition of RNA release by SNAP was less complete, with only a 55% inhibition at 8 mM of the NO donor.

In addition to demonstrating the effects of SNAP, Figure 6A–C, highlights the massive amount of mitomycin-induced nucleic acid released from E_clo_Niagara in these experiments. The combined concentrations of nucleic acids (dsDNA + ssDNA + RNA) released into the extracellular medium were quite large, totaling 141 µg/mL after a mere 2.5 h of exposure to the drug.

Since we had been surprised before as to the extent of the phenotype we were studying, we again extended our measurements to determine if protein was also released from bacterial cells by SOS inducers. Figure 6D shows that large amounts of protein were also released by mitomycin C treatment, and the protein release was also inhibited, at least partially, by SNAP.

Last, we tested the inhibitor of the bacterial generalized stress response, dequalinium [9], for its effects on DNA release. As shown in Figure 6E, 20 µg/mL dequalinium HCl partially inhibited both bleomycin- and mitomycin-induced dsDNA release. At a higher concentration of 60 µg/mL, dequalinium was able to achieve an almost complete reversal of mitomycin-induced DNA release. The results of Figure 6 suggest that both nitric oxide donors and dequalinium have promising activity as inhibitors of the SOS response in general and of SOS-induced nucleic acid release in particular.

## 4. Discussion

Our previous study on SOS-induced DNA release was intriguing, but it turned out to be incomplete [2]. In our current study, we have extended our understanding of the DNA release phenomenon, but our results raise many new questions as well.

Based on our results comparing the wild-type STEC strain EDL933 with its ΔrecA mutant, EDL933R, the DNA release triggered by the SOS-inducing drugs is completely dependent on the presence of RecA. We believe the DNA release is a previously unrecognized attribute of the SOS response itself, rather than an off-target drug effect or other non-specific action. We are unsure why the SOS-induced DNA release was not discovered long ago. One clue may come from the results in Figure 2C–G, in which the presence of capsule appeared to block DNA release. Capsule-expressing strains, such as *E. coli* MG1655, which has the K-12 capsule, have commonly been used in SOS research. An additional factor may relate to the fact that highly passaged laboratory strains can become “domesticated”, with an SOS response that is blunted compared to wild-type strains [32,33]. A third factor is that the strength of the SOS response does vary from strain to strain. For example, the SOS response appears to be especially robust in STEC [34], the so-called “hair-trigger” induction. A fourth, and technical, factor at play may be the shorter duration of exposure to the SOS-inducing drugs that we used. We have noted that, with some strains and at later times of incubation, the DNA release shows a paradoxical decrease. We think this may be because, at later times, more DNA becomes enmeshed in the bacterial pellet or deposited on the wall of the tube (Figure 4), reducing the DNA measurable in the supernatant.

Our previous report on DNA release was incomplete in that we assumed that the nucleic acid being released in response to SOS inducers was all double-stranded DNA (dsDNA). Indeed, dsDNA is released, but ssDNA is released in greater amounts. As shown in Table 2, ssDNA exceeded dsDNA by a factor of 4.7 (average of 5 species and strains; see also Figure 5 and Figure 4). As stated above in Results, the total concentration of SOS-induced nucleic acids released in the culture supernatants exceeded 100 µg/mL, at least in the most robust responders (*E. coli* EDL933 and *E. cloacae*). In addition to the release of nucleic acids, large amounts of protein were also released into the supernatant medium (Figure 6D), which may be important since protein and DNA are present together in biofilms (Figure 4B–F).

Crane and Catanzaro showed that extracellular dsDNA released by SOS-inducing drugs was incorporated into large bacterial aggregates [2]. We thought that the bacterial clumps might be precursors of biofilms, and now we know that DNA is indeed incorporated into biofilms (Figure 4). The role of extracellular dsDNA in biofilms has been well-studied [35].

Most of the extracellular DNA detected in biofilms has been of small size, which is in agreement with what we observed (Figure 5E), where much of the dsDNA was in the range of 400 to 700 bp. In contrast to double-stranded DNA, reports of ssDNA being incorporated into biofilms have been much fewer [36]. RNA has also been shown to be incorporated into biofilms in some reports [37].

Interestingly, the extracellular RNA in biofilms has been even smaller in size, usually 15 to 40 nucleotides, and typically derived from non-coding RNAs, such as tRNA and other non-coding small RNAs (ncRNA). Some pathogens are able to actively secrete RNA into the extracellular space, including in membrane vesicles [38]. Other authors noted that induction of the SOS response promoted biofilm formation [5,27], but did not appreciate that the SOS response itself triggers release of extracellular DNA that can be then incorporated into the biofilm. Extracellular DNA (eDNA) is found commonly in the natural world, but the mechanisms of DNA release have not been clearly determined in most cases [39]. Our current study, which focused on DNA release induced by drugs, showed that the SOS response was clearly involved (Figure 1) and includes a role for RecA protein (Figure 4D,E), but does not indicate the route by which nucleic acids exit the bacterial cell. The Type IV secretion system, best studied in *Agrobacterium tumefaciens* [40], is an efficient exporter of DNA, but is not present in the strains that we have used.

We and others have been interested in finding drugs that could inhibit the SOS response [9,41,42,43]. Therefore, we again tested inhibitors of the SOS response that we had previously identified, including zinc acetate and the nitric oxide donor S-nitroso-acetyl-penicillamine (SNAP), to see if they would also inhibit nucleic acid release. Zinc inhibited the release of all the nucleic acids that we had detected (dsDNA, ssDNA, and RNA; Figure 5), while copper had weaker inhibitory activity. SNAP also inhibited mitomycin C-induced release of all three nucleic acids (Figure 6). Dequalinium, a quaternary ammonium compound identified by the Rosenberg laboratory as an inhibitor of the general stress response in *E. coli* [9], also showed remarkably strong ability to inhibit SOS-induced DNA release. Our results, together with those of Zhai et al., suggest that the SOS response may finally be on the verge of being considered a “druggable target” to prevent emergence of new antibiotic resistance, as well as possibly blocking biofilm formation at the same time.

We know that the major pharmaceutical companies have lost interest in developing new antibiotics because they do not consider antibiotics as producing the same return on investment as other drug classes. Since this is the case, drugs targeting the SOS response may have to be sought from existing drugs that could be repurposed. One thing that zinc, nitric oxide donors, and dequalinium have in common is that they are already in clinical use in human medicine. In the case of the nitric oxide donors, the most relevant drugs are the oral nitrates isosorbide mononitrate and isosorbide dinitrate, which release NO slowly in the GI tract. This is fortunate because the GI tract is often the location in which antibiotic resistance first emerges [44]. Use of dequalinium would also be limited to application on mucosal surfaces since it is too toxic to administer parenterally. Dequalinium is approved by Health Canada as a vaginal suppository for bacterial vaginosis, and it is approved in the United Kingdom, India, and other countries as an oral lozenge.

Another feature that zinc, NO donors, and dequalinium share is their low potency, often requiring concentrations in the millimolar range to achieve inhibition of the SOS response. We previously showed that zinc and SNAP had additive inhibitory effects [8]. If this is also true of other SOS inhibitors, combination therapy might be a useful future strategy. The low potency of the SOS inhibitors would be less important, however, if they were to be applied to inanimate objects rather than administered to humans or animals. For example, they could be incorporated into coatings to prevent biofouling and biofilm formation [45].

In summary, SOS-inducing drugs trigger a large release of nucleic acids from Gram-negative bacteria at concentrations below or close to their minimum inhibitory concentrations (MICs). In contrast, many other antibiotics fail to trigger any nucleic acid release or only release small amounts. In response to the SOS-inducer mitomycin C, the nucleic acids released from E_clo_Niagara comprised about 58% ssDNA, 23% RNA, and 18% dsDNA. SOS inducers triggered formation of biofilm on glass surfaces, and dsDNA was a component of the biofilms. SOS-induced nucleic acid release and protein release were inhibited by zinc salts, the NO donor SNAP, and dequalinium. In addition to triggering emergence of new antibiotic resistance via hypermutation, the SOS-induced biofilm formation may also affect resistance by reducing penetration of drugs into the biofilm [6], by decreased metabolic activity of bacteria due to limited oxygen or nutrients, and by many other factors.

For many years, professionals in the fields of clinical microbiology and antibiotic stewardship have tended to ignore the SOS response as a quirk of molecular biology, of interest to those who are experts in the biochemistry of DNA repair. But as the roles of the SOS response expand [46] and the list of SOS-inducing agents increases far beyond antibiotics [47,48], understanding of the SOS response will need to be incorporated into antimicrobial stewardship efforts.

The additive effects of classical SOS inducers, such as mitomycin C, and “atypical” inducers such as fluoxetine (Appendix A), may also be important. Other authors have pointed out that sewage treatment plants are a location where many different chemicals may combine to produce “ecotoxicity” [49,50]. Wastewater effluent from hospitals and pharmaceutical plants would be even more enriched in these drugs, including antibiotics, anti-cancer drugs, psychoactive drugs, and others [51], resulting in an environment ideal for inducing new mutations conferring antibiotic resistance, followed by selection pressure to increase the fitness and prevalence of those drug-resistant strains. This implies that clinicians and antimicrobial stewardship experts may need to broaden their concept of where new antibiotic resistance occurs, as it may not be just in an individual patient or even an intensive care unit (ICU), but in the outdoor environment as well.

## Figures and Tables

**Figure 1 biomolecules-14-00321-f001:**
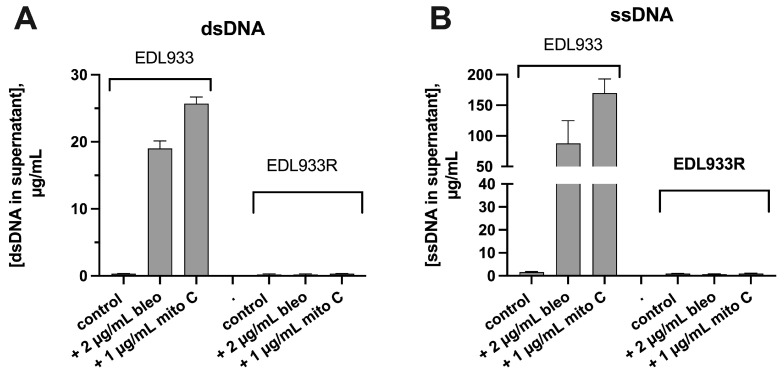
Effect of ΔrecA mutation on DNA release in STEC strain EDL933. The wild-type EDL933 and its isogenic ΔrecA mutant, EDL933R, were compared as to DNA release in response to SOS-inducing drugs blemycin (bleo) and mitomycin C (mito C). Bacteria were subcultured as described in Materials and Methods, and inducing drugs were added at 1.5 h. Medium was collected at 4 h, centrifuged, and assayed for double-stranded and single-stranded DNA. (**A**), release of dsDNA; (**B**), release of ssDNA.

**Figure 3 biomolecules-14-00321-f003:**
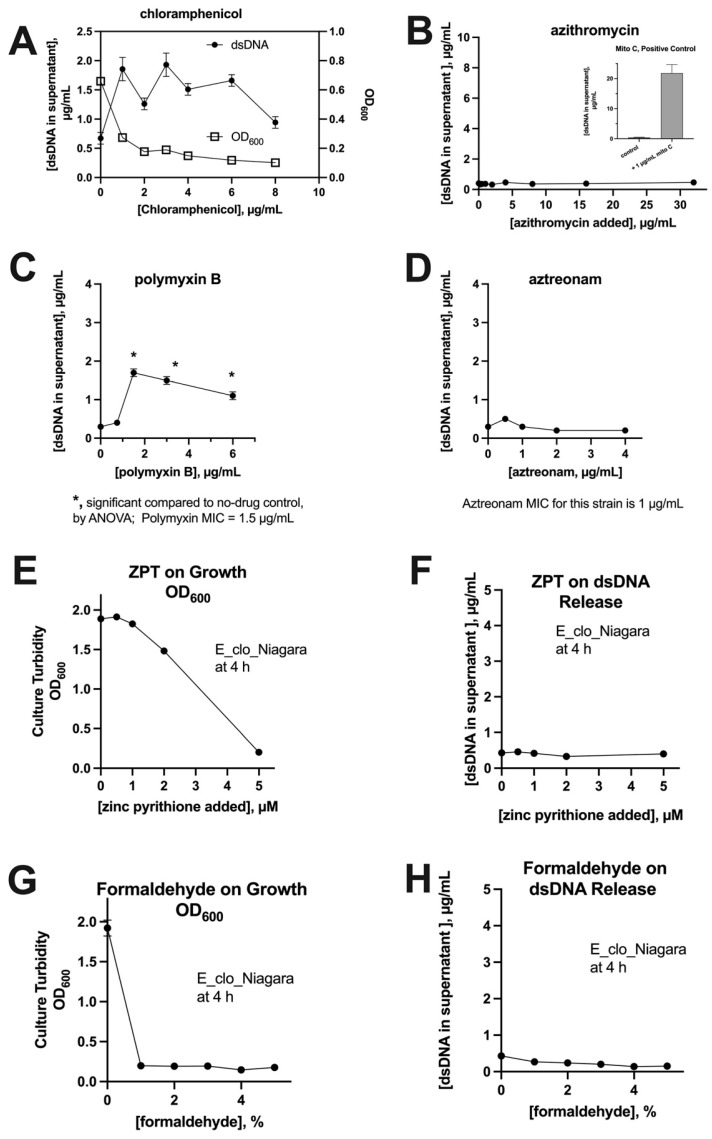
Comparison of various antibiotics for ability to induce release of dsDNA. Four antibiotics reported to not be inducers of the SOS response were tested for ability to trigger release of dsDNA from *E. cloacae* strain E_clo_Niagara. Panel (**A**): chloramphenicol, where the Left *Y*-axis shows dsDNA and the Right *Y*-axis shows OD_600_; Panel (**B**): azithromycin, with inset showing DNA release by mitomycin C; Panel (**C**): polymyxin B; Panel (**D**): aztreonam. Panels (**E**,**F**): effect of zinc pyrithione (ZPT) on growth and lack of effect on dsDNA release; Panels (**G**,**H**): effect of formaldehyde on growth and lack of effect on DNA release.

**Figure 4 biomolecules-14-00321-f004:**
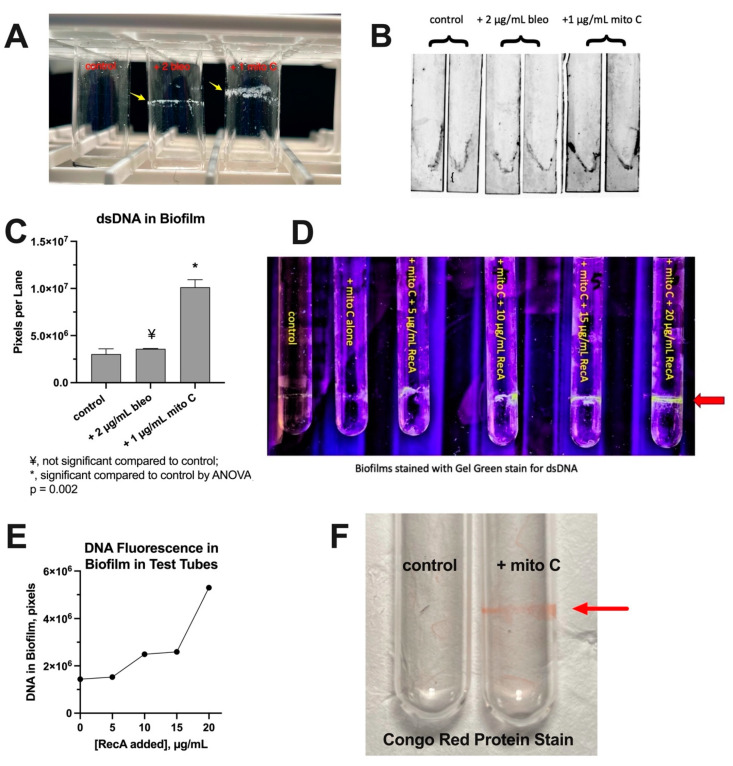
Formation of biofilm in response to SOS-inducing drugs in E_clo_Niagara. The duration of the incubation was prolonged to 4.5 h in these experiments to allow more time for biofilm to form on glass. Panel (**A**): photograph of glass test tubes showing formation of biofilm on the glass at the air–fluid interface. Untreated control tube is on the left; tube treated with 2 µg/mL bleomycin is in the center, and the tube receiving 1 µg/mL mitomycin C is on the right. Yellow arrows indicate biofilm visible to the naked eye. Panel (**B**): fluorescence image of biofilms formed on glass slides grown in 16 × 100 mm test tubes. The glass slides were air-dried, rinsed, then stained with SybrSafe dye for dsDNA, as described in Materials and Methods. Then, the slides were imaged in the GelDoc EZ fluorescence imager, and the image was “inverted” to display dark bands on a light background. With 300 rpm shaking, the liquid medium forms a vortex, resulting in V-shaped bands. Panel (**C**): quantitation of the DNA in the biofilms shown in Panel (**B**). Panel (**D**): effect of exogenous RecA protein on biofilm formation. E_clo_Niagara cultures were left untreated, treated with mitomycin C alone, or treated with mitomycin with increasing concentrations of RecA protein. The biofilms formed were stained with Gel Green stain for dsDNA, and the fluorescence was visualized under UV light (red arrow). The fluorescence images were again analyzed using the GelDoc EZ instrument, and the intensity of the mitomycin-induced fluorescent bands are shown in Panel (**E**). Panel (**F**): biofilms were stained with 50 µg/mL Congo Red dye, a protein stain, rinsed, and then photographed, showing that the biofilms contained protein as well as DNA.

**Figure 5 biomolecules-14-00321-f005:**
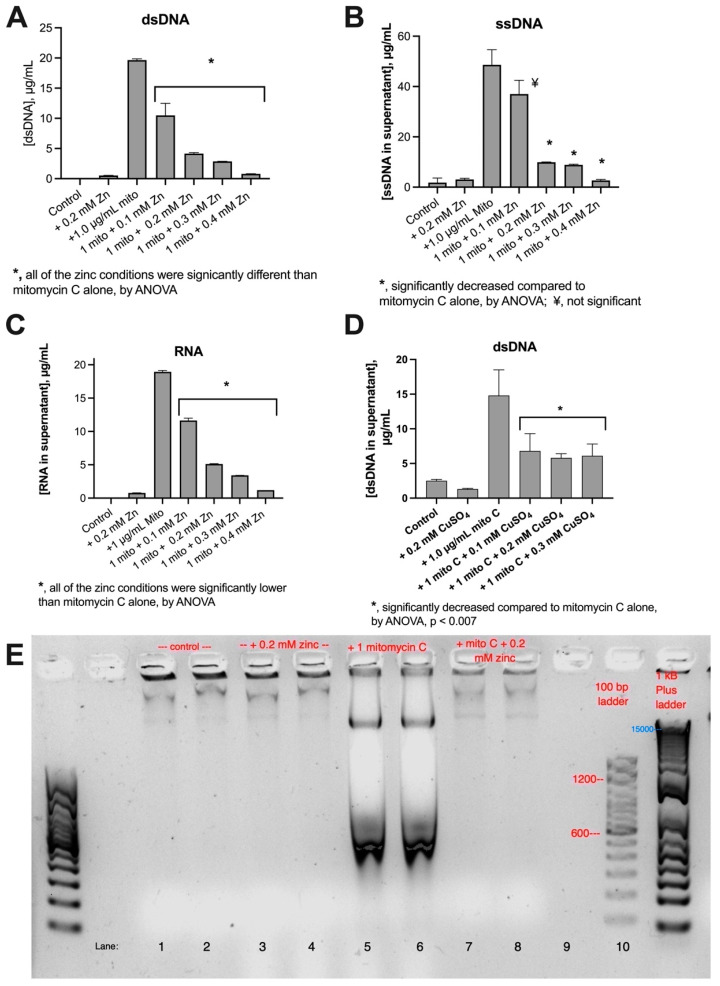
Divalent transition metals as inhibitors of SOS-induced DNA release in E_clo_Niagara. Panel (**A**): effect of zinc acetate on dsDNA release. Panel (**B**): effect of zinc on ssDNA release. Panel (**C**): effect of zinc on RNA release. Panel (**D**): effect of CuSO_4_ on dsDNA release. Copper sulfate also inhibited ssDNA release. Panel (**E**): DNA agarose gel, stained with SybrSafe dye, showing differential effects of zinc acetate on lower vs. higher molecular weight dsDNA bands released in response to mitomycin C. Image was inverted to display dark bands on a light background. DNA released by mitomycin C shows a characteristic W-shaped artifact (Lanes 5 and 6).

**Figure 6 biomolecules-14-00321-f006:**
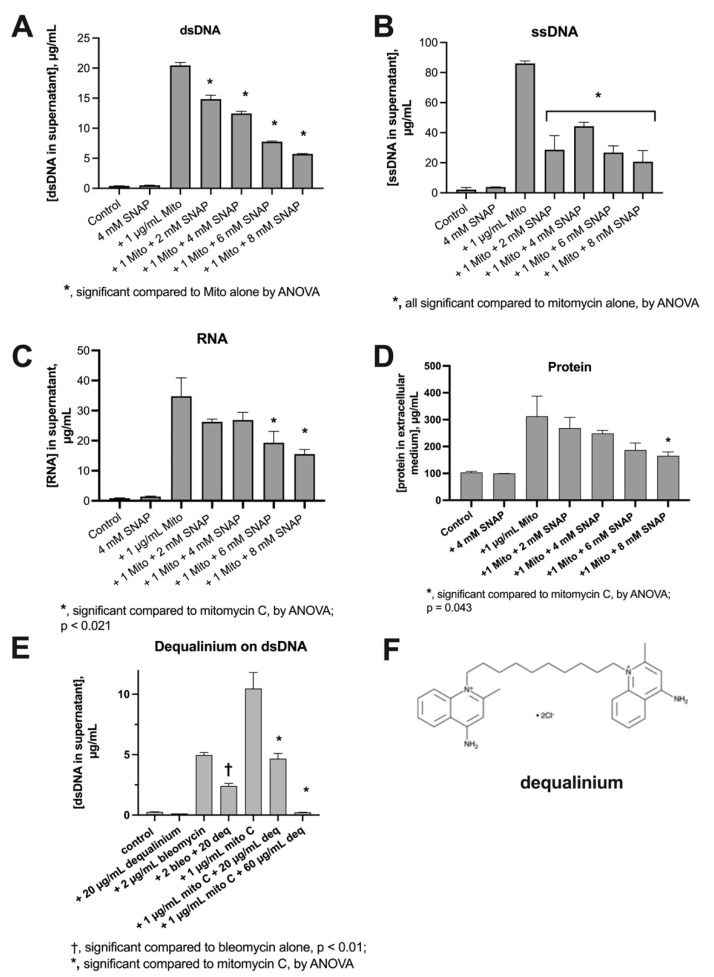
Effect of nitric oxide donors and dequalinium on SOS-induced DNA release in E_clo_Niagara. Panel (**A**): effect of S-nitroso-acetyl-penicillamine (SNAP), an NO donor, on release of dsDNA. Panel (**B**): effect of SNAP on ssDNA release. Panel (**C**): effect of SNAP on RNA release. Panel (**D**): effect of SNAP on protein release. Panel (**E**): effect of dequalinium, an inhibitor of the bacterial generalized stress response, on release of dsDNA. Panel (**F**): chemical structure of dequalinium, a quaternary ammonium compound. In addition to demonstrating the effects of SNAP, (**A**–**C**), highlights the massive amount of mitomycin-induced nucleic acid released from E_clo_Niagara in these experiments. The combined concentrations of nucleic acids (dsDNA + ssDNA + RNA) released into the extracellular medium were quite large, totaling 141 µg/mL, after a mere 2.5 h of exposure to the drug.

**Table 2 biomolecules-14-00321-t002:** Comparison of mitomycin C-induced release of ssDNA vs. dsDNA in 5 strains of enteric coliform bacteria.

Species and Strain	Comment	Ratio of ssDNA to dsDNA *
STEC EDL933	Mean of 3 experiments	6.8
EPEC E2348/69		4.5
*E. coli* Iceberg		5.1
*E. cloacae* E_clo_Niagara	Mean of 11 experiments	5.6
*Klebsiella aerogenes*		3.9
		Mean Ratio of 5 strains: 4.7 ± 1.3, *n* = 5

* all strains were treated with 1 µg/mL mitomycin C as the SOS-inducing drug.

## Data Availability

We have no additional data beyond what are included in the manuscript and the Appendix A.

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
