# Peer review of "SOS-Inducing Drugs Trigger Nucleic Acid Release and Biofilm Formation in Gram-Negative Bacteria"

_biomolecules, 2024, doi:10.3390/biom14030321_

Round 1
Reviewer 1 Report
Comments and Suggestions for Authors
This study is follow up on the authors' previous study, which further characterizes the phenomenon of DNA release from SOS induced Gram negative bacteria. The work is well done and nicely described.
The following suggestions are for the authors' consideration for their perspective work. The difference in SOS induction signal that causes the release of DNA, RNA and proteins (e.g. mito C in Pseudomonas aeruginosa) suggests that the induction of SOS response per se is not enough for the release of macromolecules from the cells. The trigger seems to be impaired replication. What about the UV induction of SOS? Also, there is a mutant lexA71 allele in E. coli, which enables constitutive SOS induction, as well as some recA mutants that render SOS induction thermoinducibe. Using those mutants may help in further elucidating the trigger for the phenomenon.
I wonder if the mechanism for DNA release is simply due to cell lysis, since it includes massive amount of various macromolecules. The effect of polymyxin B argues in favor of that explanation. I am aware that the authors previously failed to detect GFP protein release from the cells, but this study did show protein release from SOS induced bacteria. Could you follow bacterial population during SOS-inducing stress and determine CFU titer as well as visualize them by microscope? Maybe you will be able to notice a fraction of lysing cells.
Furthermore, biofilm is formed in SOS-induced cultures, which coincided with the release of macromolecules. It would be interesting to use supernatant from the SOS-induced populations, containing released macromolecules, and grow the (isogenic) SOS-uninduced bacteria in that medium. Will they develop biofilm?
Author Response
Response to Reviewer 1

Reviewer 2 Report
Comments and Suggestions for Authors
Reviewer comments
The manuscript provides interesting findings concerning the role of SOS-inhibitory drugs against bacterial DNA release and biofilm formation. The authors achieved well the results which clearly demonstrated that probiotic films aided prevent and treat antibiotic-resistance pathogenesis. Therefore, I recommend the publication of this manuscript after a major revision would be done. Comments are presented below:
-The abstract mentioned reference not properly formatted. I suggest to follow standard format this section (JK Crane & MN 17 Catanzaro, Antibiotics 2023, 12, 649) need to revise, and all the bacterial organisms need to follow uniformly with italics.
- Lines 63-65 “We realized that our previous report………. SOS-dependent” missing their supporting reference.
-Materials and Methods section need to be improve and need to be explain with under sub-titles.
As an example, Chemicals, bacterial culture, maintenance and treatment procurement procedure…etc.
-DNA, RNA isolation, protein assays, antibody staining protocols and statistical analysis statement need to explain elaborately.
-Figure 1 and 2 have to move under the results part, and the present fig. 1 legend should be move to under materials and methods. Further need to be correct the figure 1 legend as per journal requirements.
-Table 1 need to simplify and need add supporting reference for all bacterial strains.
-Results part not clear, the authors followed like review articles, it should need to revise with subheading and should avoid the previous studies supporting statements. So, better to be move those general points under the discussion section.
-All the figures with inside frame, the author mentioned significant statement, actually its need to move figure legends.
-missing authors contributions and other statements…..etc.
-All the reference not in uniform, need to be reformatting and follow with the journal guidelines.
Author Response
Response to Reviewer 2
